# Using Date Palm Waste as an Alternative for Rockwool: Sweet Pepper Performance under Both Soilless Culture Substrates

**DOI:** 10.3390/plants13010044

**Published:** 2023-12-22

**Authors:** Muein Qaryouti, Mohamed Osman, Abdulaziz Alharbi, Wim Voogt, Mohamed Ewis Abdelaziz

**Affiliations:** 1The National Research and Development Center for Sustainable Agriculture (Estidamah), Riyadh Techno Valley, Riyadh 11422, Saudi Arabia; 2Plant Production Department, College of Food and Agriculture Sciences, King Saud University, Riyadh 11451, Saudi Arabia; 3Business Unit Greenhouse Horticulture, Wageningen University and Research, 6708 PB Wageningen, The Netherlands; 4Department of Vegetable Crops, Faculty of Agriculture, Cairo University, Giza 12613, Egypt

**Keywords:** *Capsicum annum*, greenhouse, soilless culture, water use efficiency

## Abstract

The degradation of soil quality due to environmental conditions and improper management practices has caused a shrinkage in land areas suitable for crop cultivation. This necessitates a transition towards soilless culture systems, which offer desirable conditions for crop growth and development and increase resource use efficiency. One of the growth-limiting factors in soilless culture systems is the type of growing substrate. The use of more sustainable resources and environmentally friendly growing substrates is a challenge that affects the soilless culture industry. This work evaluates the efficacy of date palm waste (DPW) and rockwool as growing substrates for sweet pepper (*Capsicum annuum* L.) under greenhouse conditions. The plant height, stem diameter, average total leaf area, φPSII, and Fm′ of leaf fluorescence show significant increases when plants are grown in rockwool. No differences are found in terms of the total yield or the number of marketable fruits and fruit quality between the two substrates. However, the DPW substrate shows a significant decrease in the number of unmarketable fruits and number of Blossom End Rot (BER) fruits. Plants grown in both growing substrates consume equal water amounts for the optimal fruit production, while the water use efficiency of rockwool is better than that of DPW. Our results highlight DPW’s role in soilless production and as a key solution for resource-saving production systems.

## 1. Introduction

Intensive vegetable production requires the development of new greenhouse strategies to cope with the increasing global population demand and promote food security [1,2]. Soil degradation and low water and nutrient use efficiencies have become a global challenge, particularity in arid climate zones, which has stimulated the expansion of the use of soilless cultivation methods. Soilless culture is a crop production method that uses either mineral or organic substrates as a growing medium in greenhouse crop cultivation [3]. The soilless culture technique is suitable when there is a water shortage [3] and low soil fertility [4], and it enhances plant growth under abiotic stress factors, mainly salinity and drought [5]. Nevertheless, soilless culture involves a controlled root system and a diminished root-zone volume in comparison to soil production [6]. Moreover, the growing substrate is a key component in soilless production systems, and it can influence production and the quality of the crop. Therefore, the selection of a high-quality growing substrate is an important factor that needs to be considered to establish an efficient soilless crop production system and to maximize the yield potential. The selection of soilless substrates is based on their cost, formation properties, and ability to supply crops with sufficient water, nutrients, and air in the root zone [7,8]. However, the choice of an appropriate growing medium in soilless culture largely depends on its desirable characteristics, such as water uptake, retention, and release ability; high cation exchange capacity; minerals; and trace element retention ability [9]. For instance, rockwool is considered the best inorganic-based medium for soilless culture, with no proper waste processing at the end of the growing cycle [10]. Perlite also has the capability to act as a soilless medium due to its unique properties, despite some difficulties [11]. In addition to rockwool and perlite, several reports have indicated that volcanic rock and vermiculite are commonly used growing media in soilless culture [12,13]. On the other hand, numerous organic raw materials have been used as soilless substrates for vegetable production, such as rice hull, sawdust, coconut fibers, and peat moss [14], which enables countries with limited industrial resources to use available organic residues as alternative growing substrates. In this respect, in areas where on-substrate vegetable cultivation is greatly practiced, growers try to implement local factory-made substrates [15]. Thus, there is a growing need in many parts of the world to find alternative growing substrates that are locally available, sustainable, economically viable, and environmentally friendly [16]. Consequently, the identification of cheap and locally produced organic and/or inorganic soilless growing media is a major challenge for protected culture crops [17]. Saudi Arabia is among the leading producers of dates, accounting for 17% of global date production. Due to its suitable climate conditions for the growth of the date palm tree, it has over 31 million palm trees, with annual production exceeding 1.5 million tons [18]. Meanwhile, date palm cultivation generates an enormous quantity of waste, most of which is disposed of through incineration [19]. In [20], the authors reported that over two hundred tons per year are created from the pruned branches of palm trees in Saudi Arabia, while more than 400 tons of waste are created globally from palm trees [21]. Date palm waste (DPW) raw materials include leaves, branches, stem barks, and fronds generated during the annual trimming of palm trees. A proper method for the recycling of DPW must be sought, including its use as a growing medium in soilless crop production systems. DPW is abundantly available at a low cost, while its physical and chemical properties vary according to the date palm cultivar and growing conditions. In addition, the particle size of ground DPW can also influence its physical and chemical properties [22], thereby affecting its function when used as a growing substrate. Research regarding the potential use of DPW as a growing substrate in soilless vegetable cultivation systems is still limited, and further investigation is required [23].

Pepper (*Capsicum annuum* L.) is an important Solanaceae greenhouse crop used worldwide for domestic and commercial purposes, since it is a good source of natural pigments and antioxidants [24,25]. In general, it is ranked as the third crop in terms of cultivated greenhouse area, yield production, and the relative importance of greenhouse crops, after tomatoes and cucumbers, in Saudi Arabia [26]. Therefore, the goal of this work was to evaluate the effects of using local DPW as soilless-based substrate media for sweet pepper growth, yield, and fruit quality under controlled greenhouse conditions, in comparison with rockwool as a the most marketable soilless substrate.

## 2. Materials and Methods

### 2.1. Date Palm Waste Preparation

As previously described by [27], date palm waste (DPW) was collected from a local farm near the Riyadh region and ground to particle sizes of 8–10 mm. Briefly, after grinding, fractions were cleaned with reverse osmosis water to remove any salt build-up for one week; then, they were sterilized at 120 °C for one hour. Packaging was performed in black–white polyethylene slabs (100 × 20 × 10 cm); each slab contained approximately 4 kg of material (Figure 1).

### 2.2. Basic Characteristics of Date Palm Waste

The water holding capacity (%) for the rockwool slabs (100 × 20 × 10 cm, Gorodan, Netherlands) and DPW substrates was estimated as the difference between the wet masses (WM) and dry mass (DM) divided by the original water-saturated (WS) sample volume [WHC = (WM − DM)/WS × 100]. Organic matter was determined using the wet digestion method according to [28]. Bulk density was determined according to [29]. The cation exchange capacity (CEC) was measured according to [30]. The pH and EC values for all media before planting were determined by a pH and EC meter. 

### 2.3. Plant Material and Growth Conditions

Seeds of sweet pepper (*Capsicum annuum* L.) cv. “Clavesol (Rijk Zwaan company, Enkhuizen, The Netherlands) were surface-sterilized in 70% ethanol for 1 min, submerged in 5% NaClO for 10 min, and washed with sterile water. Sterilized seeds were divided into two groups, one placed in trays filled with potting soil (Sphagnum peat moss, Turba Rubia 5–20 mm, UK) and the second in cubes of rockwool for germination. Experiments were conducted at the National Research and Development Center for Sustainable Agriculture (Estidamah), Riyadh, Saudi Arabia (46037′ E, longitude and 24039′ N, latitude), in the middle of January during the 2019 and 2020 growing cycles. In the first week of February, seedlings with four true leaves were transplanted into the greenhouse-controlled glasshouse, a Venlo type, with 22–25/18–20 °C day–night temperatures and 75% relative humidity, till the end of the experiment, in the first week of September. Plants were placed on slabs filled with rockwool, as an inorganic substrate, and processed with the shredded pruned leaves of date palm (*Phoenix dactylifera*) as an organic substrate. Four seedlings per slab were planted at a 25 cm distance from the pre-formed wholes, while the plant spacing was 1.6 m between rows (36 m long), and the final plant distance was 2.5 plants per square meter, with three stems per plant. The air temperature, relative humidity, and photosynthetically active radiation (PAR) (PAR: 0–2000 μmol·m^−2^·s^−1^) inside the greenhouse were measured by three sensors (Horti-Max model MTV) placed in three different locations inside the glasshouse [27]. Plants were arranged in a randomized block design in three replications. A test of the normality of value distribution was conducted according to [31] using the SPSS v. 17.0 computer package.

### 2.4. Nutrient Solution

Plants were daily fertigated with nutrient solution prepared according to [32]. The nutrient solution was diluted automatically by using fixed dilution factors based on the continuous measurement of the electrical conductivity (EC) (2.5 dSm^−1^) and pH (5.5–6.0). The composition of the concentrated solution was adjusted according to the water quality and crop growth stage. The concentrated solutions were split into two tanks, A and B, where, in tank A, all Ca andNH_4_ fertilizers were added, and, in tank B, all SO_4_, H_2_PO_4_, and Mg fertilizers were added to avoid precipitation. Fe chelates were added to tank A and all other micronutrients to tank B, according to [27,32] (Table 1).

### 2.5. Macro and Micro-Elements in the Soilless Media

Nitrogen was determined by using an elemental analyzer (Leco CHN-600). Briefly, an air-dried sample (0.15–0.25 g) was added to a Leco #501-059 crucible and combusted at a high temperature (950 °C) in pure oxygen. The gases produced were analyzed for CO_2_ by a selective infrared detector, and an aliquot was obtained for the thermal conductivity detection and nitrogen measurement [33]. The Leco CHN-600 Elemental Analyzer was standardized with Leco standard #501-441 (sucrose, 40.0% C) and the National Bureau of Standards (NBS 1575, pine needles, 1.2% N). Samples were mixed with acid–perchloric acid–nitric acid for digestion, according to [34]. Potassium was measured using a flame photometer (Microprocessor 1382), while phosphorus was measured with molybdophosphoric blue [35] using a spectrophotometer (Lambda EZ 150, PerkinElmer, Waltham, MA, USA). Micronutrients were extracted using the ammonium bicarbonate–diethylene triamine pentaacetic acid (AB-DTPA) method [36] and determined according to [30] using inductively coupled plasma (ICP-OES, Optima 4300 DV, PerkinElmer Inc.) (Table 2).

### 2.6. Vegetative Growth Traits

Five plants were selected randomly from three replications at 90 days post-transplanting to measure the plant height, using a meter rule, from the base of the plant to the apical leaf. The total leaf area was calculated using a portable area meter (Li-3000A). The stem diameter was determined using a Vernier caliper at 15 cm above the ground. In addition, the number of leaves per plant was detected by counting all fully expanded leaves on the plants, leaving the bud primordial at the shoot apex. Meanwhile, the leaf and stem fresh weights were measured using an electronic balance; then, fresh samples were dried for three days at 72 °C to measure the dry weight [2].

### 2.7. Leaf Chlorophyll Index and Chlorophyll Fluorescence Parameters

A relative SPAD meter (Minolta SPAD-502) was used to measure the relative concentration of chlorophyll in the top three mature leaves. In the same leaves, leaf chlorophyll fluorescence characteristics (photosystem II (PSII), steady-state chlorophyll fluorescence (Fs), maximum fluorescence (Fm), quantum yield of photosystem II (φPSII), photosynthetic efficiency in a dark-adapted state (Fv/Fm)) were measured by the Li-6400X Portable Photosynthesis System (LI-COR Biosciences GmbH., Frankfurt, Germany).

### 2.8. Vitamin C, Titratable Acidity, and Total Soluble Solids

Two months after cultivation, five mature fruits was used to assess vitamin C and titratable acidity (TA) using the titration method, as described by [37]. Total soluble solids (Brix) were measured using a digital refractometer (PR.101 model, ATAGO, Tokyo, Japan).

### 2.9. Yield and Water Use Efficiency (WUE)

Fruits were ready to harvest from the beginning of April till the middle of September. The total yield per meter square was determined by continuous fruit harvesting at a mature stage. Representative fruit samples were dried at 70 °C to determine the fruit dry weight. Five fully colored fruits were randomly selected to measure the fruit length, diameter, and skin thickness by a digital caliper. Water use efficiency (WUE) was determined by dividing the total water applied (L·m^−2^) by the total fruit yield (kg·m^−2^) according to [38]. Water consumption was calculated as (irrigation water + cooling water − drain water).

## 3. Results and Discussion

Soilless culture is an effective system of growing plants in a nutrient solution under controlled environmental conditions. There are many agricultural substrates that can be utilized in soilless systems as an alternative for traditional growing soil. From an ecological perspective, DPW is a good alternative substrate for rockwool, which not biodegradable and carries ecological concerns [10]. The results showed significant differences between the two growing substrates in terms of their effects on plant parameters, with rockwool increasing the plant height, stem diameter, and average leaf area by 19, 21, and 36%, respectively, compared to DPW (Table 3).

Although the date palm waste did not affect the stem FW and dry matter content, the leaf FW, or the number of leaves per plant of sweet paper, a significant increase in leaf dry matter was recorded. However, the significant plant height increases of sweet pepper plants grown in the rockwool substrate reflected an enhancement in the average leaf area (1724 cm^2^) compared to plants grown in DPW (1105 cm^2^). The better growth performance of sweet pepper grown in the rockwool substrate can be associated with its physical properties (Figure 1). The authors of [39] reported that the physical properties of the growing media affect the water holding capacity and air content in the tested medium, and this affects plant growth and development.

As shown in Figure 1, the high porosity and high water holding capacity of the rockwool growing media may promote plant roots to penetrate the substrate, consequently increasing plant water and nutrient uptake, which improves plants’ vegetative growth [40]. On the contrary, [41] found that the growth of tomato plants did not differ between organic and inorganic substrates. Moreover, [42,43] found that perlite and stone pumice did not influence the tomato fresh biomass or stem diameter. Furthermore, [44] found no impact for perlite and DPW on the growth rate of strawberry plants. In cucumber, the plant height and stem diameter were substantially higher with a date palm substrate than perlite [45]. Moreover, the leaf area of tomato grown in an organic coconut coir substrate was better than for rockwool [46]. In this respect, the conflicting effects of the substrates on growth traits might be related to the type of greenhouse technology and the used covering material, the efficiency of crop management, the greenhouse microclimate, or other growth conditions besides the type of growing substrate [3,47]. The evaluation of photosynthetic pigments in the leaves of sweet pepper is a crucial tool in explaining the limited influence of substrates on the growth and performance of plants. Clearly, no changes were retrieved in the chlorophyll content (SPAD), Fs, and Fv/Fm of sweet pepper leaves (Table 4).

A few reports verify that the type of substrate does not affect the chlorophyll content in plant leaves. According to [46], coir, rice husk, and rockwool had no effect on chlorophyll in tomato leaves. Meanwhile, other authors have indicated the positive effect of organic substrates on the photosynthetic performance of plants. For instance, coconut fiber improved the leaf water potential, gas exchange, chlorophyll fluorescence, PhiPSII, Fv’/Fm′, qP, and electron transport rate of tomato leaves [48]. In contrast, the data analysis revealed a more significant increase in φPSII and Fm′ for the leaves of sweet pepper grown in rockwool than DPW growing medium (Table 3). In this respect, it could be stated that the higher water holding capacity of rockwool compared to date palm waste led to enhancements in some fluorescence parameters (Figure 1), thus increasing the absorption of nutrients in the tissues of rockwool rather than DPW. The analysis of macro- and micronutrients in the soilless substrates indicated the capability of rockwool to hold higher concentrations of plant nutrients, mainly nitrogen, potassium, magnesium, and calcium (Table 2), and enhance the translocation of these nutrients from the root zone to plant leaves and improve plant photosynthesis efficiency. The authors of [49] reported that increased potassium and phosphorus in leaves caused the useful adjustment of stomatal opening and photosynthetic metabolism. Regarding the effect of the growing substrates on the yield and fruit quality of sweet pepper plants, no variations were found in terms of fruit yield or the number of marketable fruits per plant (Table 5). A similar pattern was determined in the fruit physical characteristics and fruit juice quality parameters (Table 6). Obviously, the DPW substrate did not cause differences in growth, yield, or quality in sweet pepper plants when compared to rockwool [50]. However, some reports have revealed that many substrates, such as peatmoss or rockwool, used in soilless culture have little effect on the yield and fruit quality of tomato [51].

Our result agrees with [45,52], who obtained no variations in cucumber fruit yield or TSS and vitamin C in tomato juice when plants were grown in DPW and perlite. Moreover, [53] pointed out that using organic substrates like coconut fiber, bark, or rice husk did not induce changes in the growth, yield, and soluble solid content in tomato fruits if compared to rockwool. Recently, [10] observed that utilizing hemp fiber, as an organic alternative to rockwool, did not affect the yield and quality of tomato fruits, including total soluble solids (TSS), fruit dry weight, and mineral content. Other researchers have reported that the type of growing media has a considerable influence on the sweet pepper fruit yield. For example, [54] mentioned that the early and total yield for sweet pepper were substantially better with a rice straw substrate, while [12] noticed that the highest tomato fruit TSS value was linked to coconut fiber rather than perlite. The higher unmarketable yield in rockwool as compared to DPW could be explained based on the lower levels of calcium in the tissues of pepper leaves (Table 5). Table 5 shows the higher calcium concentration in pepper fruits collected from plants grown on DPW. Of note, sweet pepper grown in DPW produced a 45.5% lower unmarketable yield and a 38.4% decrease in the number of Bloom End Rot (BER) fruits. Similar reports have confirmed that BER is a calcium-related physiological disorder influenced by the insufficient uptake and transport of calcium through the plant organs; this deficit of calcium in the fruit tissue affects the stabilization of the cell wall and pectin and plasma membranes, causing BER symptoms in fruits [55,56]. In this respect, it can be suggested that the water holding capacity, cation exchange capacity (CEC), and nutrient content change the accessibility of mineral nutrients in the root zone [10,57]. This conclusion is confirmed by Table 2, which shows that DPW is richer in macro- and micronutrients, except for phosphorous and iron, than rockwool. This might be attributed to the higher organic matter percentage and cation exchange capacity of DPW compared to rockwool (Figure 1). However, the water consumption for both substrates was similar, since it represents the amount of water used for irrigation and water cooling through the growing cycle (Figure 2). This finding is contradictory with [54], who found that the water consumption of pepper plants grown in perlite was significantly higher than that of plants grown in rice straw.

Water use efficiency (WUE) is usually calculated based on the fruit yield given per unit of water consumed by the crop [58]. The protected culture industry is continually seeking alternative substrates that are locally available, economically feasible, and sustainable, to replace conventional growing media sources [59]. The use of date palm waste (DPW) as an organic-based substrate might be a viable option for the recycling of this accumulating organic waste and also contribute to the sustainability of local soilless vegetable cultivation systems [60]. In the current study, the use of DPW as a soilless growing media for sweet pepper presented better water use efficiency than when using rockwool, because of the lower production of unmarketable and BER fruits (Table 5). This result agrees with [61], who reported that the water use efficiency of marigold was improved when the growing medium was amended with 50% of coconut coir. However, our result is not consistent with [62], who stated that the highest crop water productivity value was produced with perlite, in comparison with the other organic tested growing media. This result might be due to the better hydrological properties of the DPW growing medium [63]. In addition, DPW mainly consists of cellulose embedded in a lignin matrix, which means that DPW has a unique bio-structure that gives it good mechanical properties [19]. Furthermore, although the water content in rockwool slabs is obviously high, water movement to plant roots may be hindered, causing a restricted root environment and a low buffering capacity for water and nutrients.

## 4. Conclusions

The current study showed that the response of plant growth parameters was better in rockwool compared to a DPW growing substrate. However, the sweet pepper yield and fruit quality were similar in both substrates. Plants grown in rockwool showed more unmarketable fruits than those grown in DPW, with no difference in marketable fruits between substrates. DPW may lead to an increase in water use efficiency. Overall, DPW has high potential to be used as a growing substrate in soilless cropping systems, and it might offer sustainability for greenhouse production by closing the circular economy and reducing the environmental impact of rockwool. Further studies with different vegetable crops are required to verify the findings of this study.

## Data Availability

Data are contained within the article.

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
