# Peer review of "Using Date Palm Waste as an Alternative for Rockwool: Sweet Pepper Performance under Both Soilless Culture Substrates"

_plants, 2023, doi:10.3390/plants13010044_

Round 1

Reviewer 1 Report

Comments and Suggestions for Authors

Line 126 - The operational scheme is not numbered.

Line 134 - Specify the potting soil, providing the exact name and specification according to the substrate declaration, as there are various variations.

Line 143 - On the operational scheme, in Figure E, only one stem is visible on pepper plants at that growth stage, not three. At what height did branching begin?

Line 147 - The experiment is designated as a randomized block design with four replications. In lines 246, 266, 297, and 303, results from samples in three replications are presented. Provide an explanation for displaying average values instead of individual ones for each growing season.

Line 149 - Why are the results not presented separately for each growing season but as the average value from two seasons (the growing season factor is excluded, and the growing period is another factor generating variability)? There is no reason not to show values for each season.

Line 150 - The methodology states that a post hoc Tukey's test was conducted, while the results (Table 3, Table 4, Table 5, Table 6) mention LSD (t-test).

Line 202 - Confusing sentence, rephrase: "titratable acidity (T.A.) of five mature fresh juice?"

Line 207 - Specify the total length of the pepper plant's vegetation and cultivation.

Line 226 - In the description of Table 3, significant differences are marked with different letters, but the table does not include asterisks as indicators of significance, with different numbers of asterisks (1, 2, and 3), commonly used to denote significance thresholds of 5%, 1%, or 0.1%. The methodology indicates an analysis only for a significance threshold of 5%, along with the mention of the least significant difference analysis (LSD), while in the results, a post-hoc Tukey's test is stated.

Line 252 - Correct the sentence; a period and capital letter follow the square brackets.

Line 253 - Same situation.

Line 270 - Again mentions LSD; the methodology indicates a Tukey's test, which is unnecessary for comparing two treatments; a t-test is sufficient.

Line 297 - Same as line 270.

Line 350 - In the description of Figure 2, significance markers with asterisks are mentioned, but Figure 2 itself lacks significance markers.

Line 399 - Different font.

Line 442 - Missing space after commas before and after the year.

Line 459 - Missing space after commas, and the year is not in bold.

Line 480 - The year is not in bold.

Line 508 - The year is not in bold.

Author Response

Dear Reviewer

Thank you for the valuable revision and good comments, the answers for your comments are in the attached file.

Reviewer 2 Report

Comments and Suggestions for Authors

The manuscript titled "Using date palm waste as an alternative for rockwool: Sweet pepper performance under both soilless culture substrate" contains interesting content, but requires corrections.

Detailed comments are provided below;

Introduction

Lines 70-81

When it comes to date palm waste, exactly how much waste is produced in Saudi Arabia every year? Why is it burned, since it is organic matter? This is inconsistent with a sustainable, pro-environmental method of waste management. Is this waste biomass composted? Should these issues be mentioned in the introduction?

2. Materials and Methods

 2.1. Date palm waste preparation

Lines 95-96

Why date palm waste was sterilized at 1000C ?

2.2. Basic characteristics of date palm waste

Figure 1 is not signed under the included photos, captions a-e are hardly legible

In addition, the treatment of date palm waste should be described in more detail in the methodology - photos a-e

2.3. Plant material and growth condition

Lines 131-146

Failure to refer to the appropriate literature. According to what methodology the authors prepared pepper seeds and so on.

How long was this experiment conducted and in what months? Was this an experiment carried out in one growing season or was it repeated in the next?

2.4. Nutrient solution

Table 1 contains a recipe for nutrient solution for soilless sweet pepper cultivation, if it is a nutrient solution used for watering peppers, should [32] be added?

2.6. Vegetative growth traits

There is no reference to literature here

Subsections 2.9 and 2.10

The authors introduced too many subsections describing experience in the methodology, which distorts the analysis of experience. Subsections 2.9 and 2.10 are unnecessary, although the content should be included in the description of the experiment

3. Results and discussion

Lines 235-242

Some selected characteristics of pepper yields were significantly better on soilless wool nutrient solution, the authors explain this by its physical properties. Did the authors report the composition of date palm waste biomass? Isn't it better to add waste that is not fresh but partially stratified or composted?

The authors introduced too many subsections describing experience in the methodology, which distorts the analysis of experience. Subsections 2.9 and 2.10 are unnecessary, although the content should be included in the description of the experiment

Comments on the Quality of English Language

minor

Author Response

(The authors gave the same response as above.)

Round 2

Reviewer 1 Report

Comments and Suggestions for Authors

Line 149:

Reviewer kindly requests the authors to provide evidence, i.e., results of a two-way ANOVA, confirming that the effect of the growing season is not significant.

The implementation of TWO-WAY ANOVA is used to determine:

  • Significance of main effects: Do levels within a factor differ alone regardless of the other factor?
  • Interaction effects: Do the levels within one factor depend on the levels of the other factor?
  • Effect size: How much variance is explained using partial eta squared from sums of squares?"

Author Response

Dear Reviewer,

Thank you for the valuable revision and good comments.

Best

Reviewer 2 Report

Comments and Suggestions for Authors

The authors responded to all comments in a comprehensive manner, so this manuscript can be recommended

Comments on the Quality of English Language

minor

Author Response

Dear Reviewer.,

Thank you for the valuable revision and good comments. 

Best

Round 3

Reviewer 1 Report

Comments and Suggestions for Authors Thanks for providing the evidence.